# Calpains at the Crossroads of Spinal Cord Physiology, Plasticity, and Pathology

**DOI:** 10.3390/cells14191503

**Published:** 2025-09-25

**Authors:** Frédéric Brocard, Nejada Dingu

**Affiliations:** Institut de Neurosciences de la Timone (UMR 7289), Aix-Marseille Université and Centre National de la Recherche Scientifique (CNRS), 13005 Marseille, France

**Keywords:** calpains, spinal cord, spinal cord injury, plasticity, spasticity, neuropathic pain, neuroinflammation, Amyotrophic Lateral Sclerosis (ALS), Multiple Sclerosis (MS), biomarkers

## Abstract

Calcium-dependent cysteine proteases, known as calpains, emerge as important regulators of spinal cord physiology, plasticity, and pathology. First characterized in the brain, they influence a wide range of processes in the spinal cord, maintaining neuronal homeostasis, shaping both synaptic and intrinsic plasticity, and modulating glial responses. When dysregulated, calpains contribute to the pathophysiology of traumatic and neurodegenerative spinal cord disorders, as well as to their associated motor and sensory complications, including spasticity and neuropathic pain. A recurring feature of these conditions is calpain-mediated proteolysis of ion channels, transporters, and cytoskeletal proteins, which promotes disinhibition and neuronal hyperexcitability. The resultant protein fragments are examined as prospective biomarkers for damage and disease progression. Meanwhile, promising strategies for neuroprotection and functional recovery in the clinic emerge as a result of innovative pharmacological and genetic approaches to modulate calpain activity. In this review, we present the current state of knowledge regarding the functions and regulation of calpains in the spinal cord and assess their translational potential as both therapeutic targets and effectors in spinal cord disorders.

## 1. Introduction

Calpains are calcium-dependent, non-lysosomal proteases found in all mammalian tissues [1]. Among the 15 members of the superfamily, calpain-1 and calpain-2 (also referred to as μ- and m-calpain according to the calcium concentration required for their activation) are the two most prevalent in the nervous system [2,3]. By fine-tuning the activity of structural and regulatory proteins through limited proteolysis, calpains support numerous physiological processes [4,5,6]. Problems arise when this activity escapes normal regulation; following trauma or in neurodegenerative disease, calpains shift from a supportive role to a destructive one, breaking down the cytoskeleton, damaging synapses and axons, stripping myelin, and, in severe cases, causing cell death [7,8,9,10].

While much of our knowledge about calpains comes from studies in supraspinal structures, their function within the spinal cord has received far less attention. This is notable, as the spinal cord forms the principal link between brain and periphery, integrating sensory inputs and motor commands, and generating rhythmic locomotor activity [11]. Because of its unique structure and function, grasping the role of calpain in the spinal cord is fundamental to understanding not only its normal physiology but also its vulnerability to injury and disease.

To this end, the present review explores calpain biology in the spinal cord. We first provide an overview of their spatial and cellular distribution, their physiological roles, and their involvement in synaptic plasticity. We then provide a description of the pathological consequences of calpain dysregulation in the spinal cord and discuss the translational potential of calpain-derived biomarkers. We conclude by examining emerging therapeutic strategies to target calpain activity and their potential to improve neuroprotection and recovery.

## 2. Calpains in the Spinal Cord

Calpain-1 and calpain-2 are the two isoforms identified in the spinal cord, and their expression has been observed in microglia, astrocytes, and neurons [12]. Their precise localization is essential for understanding their physiological functions and potential correlation with spinal pathology. Standard conditions result in the restriction of both isoforms to the intracellular environment, with no extracellular detection observed [13]. Rather than freely diffusing in the cytoplasm, calpains adhere to subcellular structures, such as the endoplasmic reticulum, Golgi apparatus, mitochondria, and cytoskeletal filaments [14,15,16,17]. In response to cellular signals, such as calcium influx, calpains can relocate to the plasma membrane, and their interaction with phospholipids allows for their activation [1,18]. At the cellular level, immunohistochemical analyses have identified calpain-1 and -2 within axons and myelin sheaths, indicating that they may play a function in the maintenance of axons and the integrity of myelin [19]. At the regional level, calpain expression is significantly higher in white matter than in gray matter, a pattern that is consistent across rodents and bovines [12]. In the ventral horn, motoneurons are significantly overexpressed in calpain-2 and can be up to ten times higher than those of other spinal cell types [20]. It is important to acknowledge that calpain-2 has the ability to restrict the proteolytic capacity of calpain-1, thereby reducing its activity [21]. On the one side this highlights a protective role for calpain-2, reinforcing intrinsic control mechanisms specifically developed to protect highly sensitive cells like motoneurons. On the other hand, its increased expression may also contribute to the increased susceptibility of motoneurons in degenerative diseases, such as ALS. Spinal interneurons exhibit substantially lower levels of calpain expression than glial cells. In accordance with their established roles in neuroimmune signaling and reactive responses to injury, astrocytes and microglia express both isoforms robustly, particularly in white matter. This establishes calpains as essential mediators of glial activation and neuroinflammation [12,19].

This heterogeneous yet overlapping distribution challenges the long-standing assumption that calpain-1 and calpain-2 share identical functions simply because they are ubiquitously expressed and structurally similar. As the following sections will show, the two isoforms can be differentially expressed in a variety of neurological disorders, including those that affect the spinal cord.

## 3. Toward a Dynamic View of Spinal Plasticity: Are Calpains the Missing Piece?

### 3.1. Forms and Paradigms of Spinal Plasticity

Previously considered a static pathway, the spinal cord is now acknowledged as a plastic structure that is capable of synaptic, structural, and functional modifications [22,23,24]. This plasticity allows the spinal cord to actively participate in motor learning, skill acquisition, and adaptation to environmental demands, thereby challenging the outdated notion that it is an inflexible entity [25,26,27]. Several experimental paradigms, summarized in Figure 1, highlight the range of spinal plasticity. One of the most powerful approaches for studying spinal plasticity is operant conditioning of reflexes. As illustrated in the top left quadrant of Figure 1, both animals and humans can be trained, using reward-based feedback, to increase or decrease the amplitude of the monosynaptic H-reflex over days or weeks [28,29,30,31,32,33]. This learning produces long-lasting modifications throughout the reflex arc, involving synaptic adjustments and intrinsic changes in motoneuron properties (reviewed in [34,35]).

The scope of spinal plasticity extends beyond operant learning, having also been demonstrated in classical (Pavlovian) conditioning. The top right quadrant of Figure 1 illustrates how repeated pairing of a peripheral nerve stimulus with a nociceptive input produced conditioned reflex responses that were NMDA-receptor dependent [36,37].

It is also possible for spinal networks to exhibit non-associative forms of plasticity. For example, habituation and sensitization of withdrawal reflexes arise from repeated exposure to the same sensory stimulus, whether benign or noxious [38,39]. This is illustrated in the bottom right quadrant of Figure 1, where the sensitivity to pain is reduced as a result of repeated stimuli. In addition, prolonged stimuli (stretching) and patterned afferent input (cyclic passive movement) can both modulate exaggerated spinal reflexes and aberrant involuntary motor output typical of spasticity in people with a chronic spinal cord injury [40].

Motor training is another factor that contributes to spinal plasticity, as evidenced by the bottom left quadrant of Figure 1. In spinalized rodents and cats, the spinal cord can re-learn coordinated stepping movements through regular treadmill training [41,42,43,44,45]. In a similar vein, treadmill training that supports body weight can significantly improve gait in people with spinal cord injury [46,47,48]. These enhancements are ascribed to the plastic reorganization within central pattern generator (CPG) circuits and adaptation at the motoneuron level [41,42,49,50,51,52], suggesting that latent spinal circuits are gradually recruited and reconfigured to produce locomotor movements. Outside of laboratory paradigms, motor practice can also induce spinal adaptation in humans. For example, professional ballet dancers exhibit significantly reduced H-reflexes in their leg muscles as a result of long-term spinal modulation resulting from intensive training [53].

### 3.2. Molecular Mechanisms of Synaptic Plasticity

Long-term potentiation (LTP) and long-term depression (LTD) at sensory and motor synapses are considered to be the substrates of motor learning and experience-dependent plasticity in the spinal cord [54,55,56]. The molecular pathways that connect neuronal activity to long-term changes in synaptic efficacy have been the subject of decades of research in the brain. It follows that activity-dependent calcium signaling is a critical catalyst for the cascade of molecular events that initiate synaptic remodeling, which in turn activates intracellular pathways that are involved in synaptic plasticity [57]. Specifically, calcium influx is facilitated by NMDA receptors, which subsequently activate a number of downstream kinases, including CaMKII, PKC, and ERK [58]. In turn, these enzymes enable phosphorylation, trafficking of AMPA-type glutamate receptors and consolidation of synaptic potentiation [59]. However, kinases are not the only players responsible for synaptic plasticity. In the “calpain hypothesis” of learning and memory [5,60], calpains have been identified as the primary link between calcium signaling and structural synaptic remodeling (see Figure 2A). As calcium enters through NMDA receptors, calpains proteolyze target neuronal substrates including cytoskeletal proteins such as spectrin, scaffolding molecules, adhesion proteins, and even glutamate receptor subunits [10]. The actin cytoskeleton is reorganized by calpain cleavage of spectrin, as demonstrated in Figure 2A for LTP mechanisms. This process facilitates the expansion of dendritic spines and the insertion of AMPA receptors, which are the hallmarks of reinforced synapses [10]. Simultaneously, the calpain-mediated proteolysis of regulatory proteins, including synGAP and SCOP, removes the constraints on ERK signaling, thereby facilitating synaptic transmission by promoting the insertion of these receptors into the postsynaptic membrane [6,61] (Figure 2A). In addition to modifying the morphology of dendritic spines and postsynaptic density, this regulated proteolytic activity also affects synaptic localization, which is indicative of stable synaptic changes [6].

Despite the well-established involvement of calpain in LTP, learning, and memory, there is currently no immediate evidence to support the role of this enzyme in spinal synaptic plasticity under physiological conditions. Many studies, on the other hand, have demonstrated that the manifestation of spinal synaptic plasticity is significantly influenced by the activation of NMDA receptors, particularly in the dorsal horn, where nociceptive processing takes place [62,63]. Furthermore, in vivo investigations have demonstrated that NMDA-dependent mechanisms facilitate the consolidation of learned responses in spinal H-reflex conditioning [36,64]. The well-established correlation between calpain and NMDA receptor activation implies that calpain may be involved in spinal synaptic plasticity.

### 3.3. Calpains and Inhibitory Synaptic Plasticity

In addition to the conventional glutamatergic form of transmission, modifications in inhibitory signaling can also induce synaptic plasticity. One of the primary determinants of inhibitory synaptic strength is the potassium-chloride cotransporter KCC2. This protein is responsible for maintaining a low intracellular chloride concentration, which enables hyperpolarizing responses to GABA and glycine [65]. Notably, KCC2 is particularly susceptible to calpains (as illustrated in Figure 2A), which results in functional impairment and, as a result, a decrease in inhibitory control [66,67,68]. Our most recent discoveries suggest that calpain may also regulate KCC2 in physiological contexts. In particular, we have observed that the selective inhibition of calpain in motoneurons results in an increase in KCC2 expression in the spinal cord. This suggests that endogenous calpain activity plays a role in the regulation of inhibitory tone. By extension, the dynamic modulation of inhibitory strength has the potential to contribute to LTP or LTD by regulating KCC2 in a calpain-dependent manner (Figure 2A).

Through mechanisms that involve calpain-mediated cleavage, neuronal activity dynamically modulates KCC2 diffusion and clustering at the membrane, thereby controlling synaptic efficacy and chloride homeostasis [69]. This is consistent with the fact that KCC2 downregulation in the hippocampus has been linked to LTP induction, potentially by improving EPSP-spike coupling through reduced chloride extrusion [70]. In contrast, the upregulation of KCC2 resulted in aberrant LTD and reduced LTP, both of which were reversible by inhibiting KCC2 [71]. Furthermore, in addition to its transport function, KCC2 plays a metaplastic role in glutamatergic synapses by regulating the structural and functional plasticity of excitatory synapses through the regulation of the actin cytoskeleton and cofilin signaling [72]. In conclusion, calpains have a dual effect on synaptic plasticity by influencing inhibitory synaptic function and sculpting excitatory synapses and the cytoskeletal structure.

### 3.4. Calpains in Intrinsic Neuronal Plasticity

While much has been written about synaptic mechanisms, the structural domains that govern intrinsic excitability, such as the axon initial segment (AIS), are also subject to experience-dependent plasticity [73,74]. The AIS is the primary site for action potential initiation. This feature depends on how the voltage-gated ion channels, such as Nav1.6 sodium and Kv7.2 potassium channels, are arranged. They are held in place by the scaffolding protein ankyrin G (AnkG) [75]. The AIS structure can be dynamically regulated in response to changes in sensory input or motor output, as evidenced by recent studies. Notably, the AIS shortening and altered intrinsic excitability in spinal motoneurons are indicative of this plasticity in response to long-term neuromuscular activity perturbations [76,77]. These modifications are likely a form of homeostatic adaptation, which enables neurons to adjust their excitability in accordance with functional needs. Figure 2B illustrates how calpains can regulate the AIS plasticity. They have the ability to cleave ankyrin G, which leads to the dissolution of the AIS scaffold and the relocation or loss of ion channels [78,79]. Accordingly, while Kv7.2/7.3 channel clusters at the AIS remain stable under normal conditions, they are internalized in a calpain-dependent manner when neuronal activity increases [80]. In addition to scaffold proteins, ion channels themselves are calpain substrates, including Nav1.6 and Kv7.2. In sodium channels, proteolytic cleavage disrupts inactivation and enhances persistent sodium currents [68,81], thereby lowering the action potential threshold and increasing excitability. Collectively, these findings, as illustrated in Figure 2B, underscore the extent to which calpains facilitate intrinsic neuronal plasticity. It appears that they are involved in the activity-dependent tuning of motoneuron excitability and, more broadly, in the modulation of spinal cord output by reshaping axonal structures such as the AIS and modifying the biophysical properties and availability of ion channels.

### 3.5. Interplay Between Neurotrophic Factors and Calpains

Numerous emerging lines of evidence indicate that spinal plasticity is influenced by a complex interaction between neurotrophic factors and calpains, an interaction that has been extensively documented in other regions of the CNS. In the spinal cord, instrumental learning tasks are linked to elevated levels of plasticity-associated proteins, including BDNF, CaMKII, CREB, and synapsin I [82,83,84], pointing to a molecular foundation for experience-driven spinal learning. Consistently, activity-based rehabilitation following spinal cord injury elevates levels of BDNF, NT-3, and NT-4 in the lumbar spinal cord, and these modifications are positively correlated with improved spinal reflex modulation [85,86,87]. BDNF, in particular, has been demonstrated to modulate the plasticity of inhibitory and excitatory synapses [88]. In rodents, BDNF markedly upregulates VGLUT2 in the spinal cord, which helps glutamatergic transmission [89]. At the same time, it enhances the inhibitory input to motoneurons by promoting the overexpression of GAD67 in the terminals surrounding the cell bodies [89]. The extensive influence of BDNF on spinal network function is further underscored by its regulation of KCC2, which also affects chloride homeostasis [90].

In the hippocampus, BDNF activates calpain-2 through the TrkB and ERK/MAPK signaling pathways, a finding that has prompted extensive research into the BDNF-calpain axis beyond the spinal cord. This cascade plays a central role in consolidating late-phase LTP, in part by driving PTEN degradation and promoting mTOR-dependent protein synthesis [6,91] (see Figure 2A). Interestingly, in Alzheimer’s disease, calpain disrupts the TrkB-BDNF pathway by cleaving the full-length TrkB receptor, generating a truncated form implicated in spatial memory deficits [92]. Although such mechanisms have not yet been demonstrated directly in the spinal cord, the molecular parallels suggest that similar pathways could contribute to long-term adaptations in spinal networks. Consistent with this view, BDNF has been shown to regulate expression of the KCC2 cotransporter [86,90], a function that may intersect with calpain activity, as calpains also influence KCC2 stability and function [66,67,68,93]. Figure 2A summarizes these converging pathways, highlighting how the neurotrophin-calpain crosstalk could support both synaptic and intrinsic forms of spinal plasticity.

In sum, while direct evidence in the spinal cord is still lacking, the mechanistic parallels with cortical plasticity suggest that calpains act by integrating activity-dependent signals. Operating likely downstream of calcium influx and neurotrophin pathways such as BDNF-TrkB, they appear capable of influencing several layers of plasticity from synaptic strength and inhibitory balance to AIS organization and motoneuron excitability. In this light, calpains may represent the “missing molecular piece” that helps complete our picture of spinal plasticity.

## 4. Breaking Down Barriers: Calpain-Driven Pathologies in the Spinal Cord

Calpain dysregulation in the spinal cord has been implicated in a wide range of diseases. Most research to date has focused on traumatic spinal cord injury (SCI), where extensive evidence links calpain-mediated proteolysis to cellular degeneration, glial reactivity, and functional decline. The role of calpains in other spinal cord diseases has been less well studied although emerging evidence suggests a broader role in progressive neurological diseases. In this review, we examine the pathological actions of calpains in SCI, with particular attention to their contribution to spasticity and pain and briefly consider other disorders in which calpain dysregulation in the spinal cord has been reported.

### 4.1. Spinal Cord Injury and Spasticity

Spinal cord injury (SCI) is a severe condition caused by a damage to the spinal cord responsible for temporary or permanent disability. During the acute phase of SCI, the initial trauma takes place followed by a complex cascade of events that lasts for several weeks and terminates with the appearance of the glial scar. At this stage, the direct trauma causes cell death, hemorrhage and ischemia, lastly contributing to the initiation of the immune response that exacerbates neurological deficits and outcomes. In the chronic phase, maladaptive mechanisms leading to excitotoxicity, loss of ionic homeostasis, cellular functional changes and network remodeling are fully established. Although astrocytic gliosis and inflammation dominate the tissue response to injury, neurons remain the principal victims of these processes, and their dysfunction ultimately drives functional complications. Spasticity is among the most common and disabling consequences of SCI, affecting up to 70% of patients within the first year after injury [94]. Clinically, it is defined as a velocity-dependent increase in muscle resistance to passive movement in individuals with motor system lesions [95]. This phenotype correlates with a hyperexcitable stretch reflex, as a result of molecular, cellular, and network modifications in the spinal cord. Understanding these mechanisms is of considerable clinical importance, as current treatments show limited effectiveness and various side effects. Here we discuss the time course of calpain activation after SCI and their role on the emergence of spasticity.

#### 4.1.1. Acute Calpain Activation and Early Pathological Events

After the initial mechanical insult, SCI triggers a rapid activation of calpains, as evidenced by the degradation products of cytoskeletal proteins that manifest within 15 min of the injury [96,97]. These levels are at their highest between 24 and 48 h [98,99]. This parallels with an intracellular overload in calcium, which reaches its peak approximately 8 h after SCI and remains elevated for up to 2 weeks [100]. Both calpain-1 and -2 are upregulated in adult rats acutely spinalized [101]. In the sub-lesional spinal cord of neonatal rodents, only calpain-1 levels increase [68]. It is intriguing that calpastatin levels increase gradually between postnatal days 5 and 10 [102], which may explain the rapid activation of calpain-1 after injury and the early onset of spasticity in neonatal models. As illustrated in Figure 3, calpain-1 activation occurs in multiple cell types, including motoneurons, astrocytes, microglia, and macrophages, immediately following acute SCI in the lesion core and adjacent spinal segments [99,101,103].

Calpain drives reactive gliosis in astrocytes. The calpain-dependent upregulation and cleavage of glial fibrillary acidic protein (GFAP) are a defining feature of this response [104,105]. A similar pattern is seen in Alzheimer’s disease, where astrocytic calpain cleaves calcineurin to generate p25, which in turn leads to sustained Cdk5 activation and gliosis [106]. The same pathway appears to be at work after spinal injury; calpain inhibition reduces p25 levels, limits Cdk5 overactivation, and promotes functional recovery [107,108]. This early phase of gliosis is primarily driven by calpain-1, as knockdown of this isoform significantly reduces lesion volume [105,109]. This time course coincides with the early glial proliferation that precedes scar formation [110]. In addition, calpains in the white matter contribute to early post-SCI demyelination by degrading myelin basic protein [111]. Thus, while reactive gliosis represents an important calpain-dependent feature of the injured microenvironment, it is largely the neuronal consequences of this glial response, loss of inhibitory control, cytoskeletal disruption, and excitability changes, that underlie long-term dysfunction.

In the immediate aftermath of injury, calpains also intensify the inflammatory response (Figure 3). For instance, they cleave pro-IL-1β to its active form in macrophages, thereby promoting a pro-inflammatory milieu [112,113]. Calpains can also be released by microglia into the extracellular space, where they can cause injury to the surrounding neurons and perpetuate a neurotoxic cycle [114]. Calpain-mediated effects may also extend to the vasculature. Although direct evidence in SCI remains scarce, studies in brain trauma and ischemia have shown that calpains contribute to blood–brain barrier disruption by cleaving cytoskeletal and tight junction proteins, thereby promoting vascular leakage and edema formation [115,116]. By analogy, similar mechanisms are likely to operate at the blood–spinal cord barrier, potentially exacerbating vasogenic edema in peri-lesional segments. As inflammation develops, additional triggers for calpain activation emerge; inflammatory cells generate reactive species such as peroxynitrite which maintains calpain activity [117]. During the acute phase of SCI, calpain hyperactivity is further exacerbated by the rapid depletion of its endogenous inhibitor calpastatin, at least in part due to its calpain-mediated degradation [9] raising the calpain/calpastatin ratio in injured segments [118]. Together, these events provide a permissive environment for persistent calpain overactivity (see Figure 3).

#### 4.1.2. Chronic Calpain Activity and Functional Consequences

In accordance with this, calpain activity is shown to be increased for weeks or even months after SCI. In rodent models, calpain-mediated protein breakdown products, like sodium channel fragments, are still detectable as far as 2 months post-injury [81], indicating sustained proteolysis. Transcriptomic profiling of laser-captured motoneurons adds further support; calpastatin expression drops by day 21, whereas calpain expression rises by day 60 after SCI [119] (see Figure 4A). More recently, elevated calpain expression have been found in lumbar motoneurons months after injury [93]. Functional experiments reinforce this view, showing that calpain inhibition reduces sodium channel proteolysis [81] and the knockdown of calpain in motoneurons restores KCC2 levels [93].

The time course of calpain activation appears to mirror the delayed onset of some chronic complications. Spasticity, as manifest by hypertonia, hyperreflexia, spasms and clonus, for example, typically emerges within weeks after SCI and gradually worsens, peaking at 1–2 months in adult rats [93,120,121]. Gene therapy to silence calpain-1 in lumbar motoneurons prevented this progression, confirming that calpain-1 remains active at later stages and drives the symptoms. A significant discovery in recent years is that calpain activity is directly linked to the hyperexcitability of motoneurons underlying spasticity [68,81,122,123].

#### 4.1.3. Mechanistic Links to Spasticity and Circuit Remodeling

Following SCI, a dramatic increase in spinal excitability is observed. Below the lesion, networks become spontaneously active by generating large and frequent bursts of activity. In addition, spinal reflexes are exaggerated and arise as long-lasting responses that decay after several seconds. This general spinal network hyperexcitability is supported by changes in motoneuron intrinsic properties and synaptic transmission [50,81,124,125,126]. We have also identified calpains as key mediators of these motoneuron alterations, acting on multiple molecular targets and ultimately contributing to the development of spasticity [68,93].

**Na_v_ channels.** Sub-lesional motoneurons present long-lasting firing in response to transient sensory or network activation, referred to as *plateau potentials*. Such a potential consists of a sustained depolarization of the membrane, with self-perpetuating spikes that continue well beyond the end of the initial stimulus [127] (Figure 4A). This kind of activity provides sustained excitatory drive by amplifying synaptic input on SCI motoneurons, ultimately generating exaggerated reflex responses typical of spasticity [121,128]. Plateau potentials are supported by persistent inward currents (PICs) that are largely enhanced after SCI [129]. In our work, we have focused on the persistent sodium current (I_NaP_) for its critical role in enabling plateaus in lumbar motoneurons [130]. After SCI, I_NaP_ is larger in amplitude than in intact conditions, driving pathological network output such as hyperactive reflexes and spasticity [68,81]. In lumbar motoneurons, I_NaP_ arises mainly from Na_v_1.6 voltage-gated sodium channels, which are densely expressed at the AIS [81,131,132]. Under normal conditions, Na_v_ channels inactivate rapidly after opening, a process ensured by an intracellular loop that acts as an inactivation gate [133]. Seminal work using pronase on squid axons demonstrated that the proteolytic cleavage of this inactivation gate abolishes fast inactivation, leading to a pronounced I_NaP_ [134]. After SCI, a similar process occurs; ƒ calpain cleaves Nav1.6 channels within their inactivation domain, generating a ~120 kDa breakdown product that serves as a biomarker of injury severity and spasticity, as we will discuss below [68,81,123]. In the absence of a functional inactivation gate, the pore remains open to Na^+^ influx after an action potential, resulting in I_NaP_ up-regulation, motoneuron hyperexcitability, and muscle spasms, as illustrated in Figure 4A.

**KCC2.** A second calpain-dependent mechanism contributing to motoneuron hyperexcitability in chronic SCI involves the loss of the potassium-chloride cotransporter KCC2 from the neuronal membrane. In physiological condition, KCC2 is essential to maintain low intracellular chloride levels by extruding chloride anions and hyperpolarizing the chloride equilibrium potential [135]. Therefore, the opening of chloride-permeable receptors, such as GABA_A_ and glycine receptors, promotes a net influx of chloride and induces neuronal inhibition. After SCI, a significant decrease in KCC2 expression is observed below the lesion, especially in motoneurons [51,90]. As intracellular chloride levels rise, activation of GABA_A_ and glycine receptors produces a weaker hyperpolarization, or even depolarization, shifting the balance toward spinal disinhibition. Boulenguez and collaborators [90] were the first to show that KCC2 downregulation drives spasticity in chronic SCI by reducing inhibitory tone. We subsequently showed that this effect is mediated by calpain-1 activation within motoneurons [68,93]. KCC2 exists in neurons in two different forms: an oligomeric form that is functionally present at the membrane of mature cells; and a monomeric form that is mainly cytoplasmic and expressed in immature neurons or in pathological conditions [136,137]. The oligomeric form has the highest sensitivity to calpain that targets the C-terminus due to the presence of two predicted cleavage sites (PEST sequences; [138,139]) causing the destabilization and internalization of the complex [90]. The localization of calpain-mediated KCC2 proteolysis is further confirmed by the loss of KCC2 immunostaining at the plasma membrane, given that most antibodies target the C-domain. Because post-translational modifications regulate substrate vulnerability to calpain [140], it is likely that dephosphorylation of the PEST motives might be responsible for calpain sensitivity of the oligomeric KCC2 after lesion [141]. Lastly, as a functional C-domain is required for KCC2 to extrude chloride [138,139], dysfunction responsible for spinal disinhibition might be a direct consequence of calpain activation after SCI. Thus, calpain is behind both hallmarks of spasticity; it supports excitatory drive in motoneurons by increasing I_NaP_ and decreases inhibitory strength via KCC2 downregulation. Both mechanisms lead to motoneuron hyperexcitability and spasticity (see Figure 4A).

**Other targets.** Beyond Na_v_ channels and KCC2, calpains may also affect synaptic proteins important for motor circuit plasticity and therefore synaptic remodeling during the chronic phase of SCI. After injury, sub-lesional motoneurons undergo massive reorganization, with changes in both inhibitory and excitatory synapses. The expression of GABA_A_ and glycine receptors is modified after lesion, both in adult and neonate models of SCI [124,125]. Inhibitory neurotransmission is also impaired in the sub-lesional spinal cord by affecting both GABAergic and glycinergic boutons onto motoneurons [50,126]. Although there is limited evidence for the involvement of calpains in these modifications, we argue that calpain-mediated proteolysis of synaptic proteins, cell adhesion molecules or cytoskeletal anchors might facilitate the structural and functional remodeling of inhibitory synapses. Lately, we also provided a role for voltage-gated potassium channels K_v_7.2 in the attenuation of spinal excitability by counteracting I_NaP_ [142]. Because K_v_7.2 channels are calpain sensitive and undergo membrane down-regulation in the hippocampus [80], we assume that they may be putative calpain substrates after SCI contributing to spinal disinhibition.

Calcium is another major player of spinal hyperexcitability and spasticity. Besides I_NaP_, persistent calcium currents supported by L-type calcium channels also underlie plateau potentials, motoneuron hyperexcitability and spasticity [129,143,144,145]. Furthermore, calpains are typically activated by high intracellular calcium, either in response to calcium influx through plasma membrane channels or by release from intracellular stores, such as the endoplasmic reticulum. L-type calcium channels allow for the entrance of calcium in neurons, contributing to calpain activation [146]. In turn, they are catabolized by calpains at the C-terminus in some regions of the brain [147], although no data are currently available in the spinal cord. Intracellular calcium overload is also induced by calcium-induced calcium release phenomena triggered by the activation of ryanodine receptors (RYR) in the endoplasmic reticulum. RYRs are highly sensitive to calpains in the skeletal muscle [148], which makes them possible calpain targets also in the spinal cord after SCI.

### 4.2. Spinal Cord Injury and Pain

Pain reflects the ability to detect noxious stimuli and represents an essential early warning system. Spinal cord lesion can disrupt this protective pathway, often resulting in hypersensitivity [149,150]. Typical main symptoms of neuropathic pain include allodynia (pain in response to a non-nociceptive stimulus) and hyperalgesia (increased pain sensitivity to a nociceptive stimulus). Several mechanisms contribute to central sensitization within the dorsal horn of the spinal cord. Among these, presynaptic terminals release glutamate in excessive amounts, while postsynaptic AMPA and NMDA receptors become phosphorylated and remain constitutively active [151,152,153]. This sustained receptor activation drives a rise in intracellular calcium, responsible for the activation of intracellular pathways that influence numerous downstream targets, calpains among them. The following sections will examine more closely how calpains contribute to dorsal network disinhibition and neuroinflammation and the way this is related to dorsal horn hyperexcitability and neuropathic pain.

#### 4.2.1. Calpain, KCC2, and Disinhibition

As in spasticity, disinhibition of spinal dorsal networks is a major contributor to the central sensitization that drives neuropathic pain. Dorsal disinhibition is mainly driven by a significant downregulation of KCC2 in dorsal horn neurons below the injury site [154,155,156], a change associated with increased pain sensitivity, such as allodynia and hyperalgesia [157,158]. Decreased KCC2 expression in the human spinal dorsal horn has also been linked to chronic pain [159]. Growing evidence shows that calpain activation acts upstream of KCC2 downregulation and the resulting disinhibition, as illustrated in Figure 4B. Peripheral nerve injury, sharing mechanisms with SCI-related central pain, have been particularly informative. A key study showed that nerve injury induces Calcium-dependent calpain activity in the spinal cord leading to the proteolytic cleavage of KCC2 at its C-terminal domain [67]. This degradation of KCC2 depolarizes the GABA_A_ reversal potential, so that GABAergic input excites rather than inhibits dorsal interneurons, promoting further excitability and pain transmission [160]. The central role of calpain in this process is further supported by intervention studies where the pharmacological inhibition or the selective silencing of calpain-1 at the spinal level prevents KCC2 breakdown and reduces neuropathic pain in rodents [67]. Similar results have been obtained with S-nitrosoglutathione, an agent that indirectly inhibits calpain by reducing peroxynitrite levels; preserved KCC2 expression, limited calpain-mediated spectrin cleavage and reduced mechanical allodynia two weeks after SCI [117].

Importantly, calpain regulation of KCC2 is not limited to the spinal cord. In hippocampal neurons, increased neuronal activity triggers NMDA receptor activation, Calcium influx and calpain-mediated cleavage of KCC2 [66,69]. This is accompanied by decreased KCC2 clustering and impaired chloride transport, allowing rapid, activity-dependent modulation of neuronal chloride homeostasis. In sum, calpain-mediated downregulation of KCC2 as a shared mechanism across different CNS regions, underlying spasticity and neuropathic pain after SCI.

#### 4.2.2. Calpains, Neuroimmune Interactions, and Modulation of Pain

Calpain may also contribute to neuropathic pain through neuroimmune interactions. After SCI, these glial cells exhibit a reactive phenotype, which is defined by the release of pro-inflammatory mediators, morphological alterations, and impaired clearance of glutamate and extracellular potassium. Astrocytes are likely to contribute to calpain overactivation and overexpression in the spinal cord, given their involvement in excitotoxicity and their integration within sub-lesional networks. Following SCI, microglia activation in the dorsal horn is largely driven by ATP released from primary afferent terminals, which binds to P2-type purinergic receptors on microglia [161,162]. The resulting tissue infiltration is due both to increased local microglia proliferation and to permeabilization of the blood–brain barrier, itself triggered by chemoattractant cytokines that facilitate the entry of circulating immune cells [163,164]. Once in a pro-inflammatory state, microglia release a huge number of signaling molecules, such as BDNF. By acting on the TrkB receptor signaling, BDNF downregulates KCC2, thereby altering chloride homeostasis in dorsal horn neurons. This shift sensitizes dorsal interneurons, heightening their responses and contributing to central hyperexcitability and persistent pain [165]. Building on these observations, it is tempting to draw a mechanistic link whereby microglia-derived BDNF activates TrkB receptors on dorsal horn neurons, triggering ERK and calpain activation, as shown in the hippocampus [6,91]. Calpain, once active, may then promote KCC2 cleavage, neuronal disinhibition and neuropathic pain, echoing mechanisms described before. Figure 4B provides a schematic overview of these neuroimmune interactions. Additional evidence comes from peripheral nerve injury models, where calpain-2 expressed in sensory neurons drives the production of pro-inflammatory cytokines, such as TNF-α and IL-6 that additionally sensitize pain pathways [166,167].

The role of astrocytes in central sensitization is less well defined. After SCI, these glial cells exhibit a reactive phenotype, which is defined by the release of pro-inflammatory mediators, morphological alterations, and impaired clearance of glutamate and extracellular potassium [168,169]. Astrocytes are likely to contribute to calpain overactivation and overexpression in the spinal cord, given their involvement in excitotoxicity and their integration within sub-lesional networks. They also synthesize, release and reuptake BDNF [170,171,172], suggesting possible involvement in the calpain-dependent mechanisms of central sensitization outlined above (see Figure 4B). Taken together, these data position calpains as regulators of the neuron-glia signaling, with a direct role in the central sensitization processes that sustain neuropathic pain after SCI.

### 4.3. Neurodegenerative Diseases

Calpains contribute to the progressive neuronal deterioration seen in spinal cord-related neurodegenerative disorders, such as Amyotrophic Lateral Sclerosis (ALS) and Multiple Sclerosis (MS). Although the underlying mechanisms of these conditions are distinct, animal models consistently demonstrate that calpain inhibition enhances outcomes, thereby substantiating its potential therapeutic relevance.

#### 4.3.1. Amyotrophic Lateral Sclerosis

ALS is a lethal disease characterized by the degeneration of upper and lower motoneurons, leading to progressive muscle weakness, paralysis, and ultimately death [173]. The etiology is multifactorial, but accumulating evidence involves dysregulation of calcium homeostasis and the resulting overactivation of calpains as central elements of its pathogenesis [174,175,176]. One important source for this calcium overload is spinal excitotoxicity, where excessive glutamate release and/or impaired clearance, drives sustained activation of ionotropic glutamate receptors [177]. In motoneurons, chronic calcium overload (often in combination with oxidative stress and mitochondrial dysfunction [178]), creates conditions for persistent calpain activation [179]. Of the isoforms, calpain-1 is recruited early in the disease, even at pre-symptomatic stages [175]. It cleaves critical neuronal substrates, including the NR2B subunit of NMDA receptors and αII-spectrin. The extensive cleavage of cytoskeletal and synaptic proteins during overactivation compromises neuronal integrity and function in motoneurons [180]. This disrupts axonal architecture and transport leading to axonal degeneration, muscle denervation and paralysis [181]. Interestingly, the phosphorylation state and calmodulin interactions of neurofilaments seem to influence their susceptibility to calpain [182,183]. Calpains have also been involved in the cleavage of DNA/RNA-binding protein TDP-43, a hallmark of ALS pathology, which adds another layer to their contribution in disease progression [184]. The truncated fragments mislocalize to the cytoplasm and aggregate, further exacerbating neuronal toxicity [185,186,187]. In addition, calpain can trigger caspase-dependent apoptosis of motoneurons and induce necrosis by disrupting lysosomal membranes and releasing cathepsin [188,189]. The fact that calpain inhibitors are neuroprotective and enhance motoneuron survival strongly supports its causal role in ALS [180].

Calpains probably interact with more than only neurofilaments and cytoplasmic proteins. They probably also interact with synaptic receptors and ion channels. Of particular note, their activity may contribute to the hyperexcitability of motoneurons, an early and defining feature of ALS, which is associated with increased persistent sodium currents (I_NaP_) [190,191,192,193]. This is consistent with the mechanisms that have been described in the context of spasticity following SCI. In this case, I_NaP_ is upregulated by the calpain-mediated cleavage of Nav1.6 channels [190,191,192,193]. Consistently, altered Nav1.6 expression at the axon initial segment (AIS) has been observed in ALS [194,195]. In addition to its prevalence in both patients and animal models [141,196], spasticity may also be linked to chloride dysregulation, as the progressive loss of KCC2 from motoneuron membranes has been observed in ALS models [141,197]. While there is no direct evidence of calpain-dependent KCC2 cleavage in ALS, the similarities to SCI are remarkable, as this cleavage contributes to spasticity and induces KCC2 downregulation [68,93,122].

#### 4.3.2. Multiple Sclerosis

Multiple sclerosis is an autoimmune disorder in which T cells target myelin sheaths, producing inflammatory demyelinating lesions in the CNS, often involving the spinal cord. The loss of saltatory conduction that follows leads to axonal degeneration and, ultimately, symptoms such as visual disturbances, impaired motor control and coordination, and limb weakness [198,199]. Unlike ALS, which is driven primarily by glutamate excitotoxicity, MS pathogenesis is due to T-lymphocyte activation supporting the immune response against myelin proteins [200]. There is strong evidence that excessive calpain activity is a major contributor to tissue damage in MS. First, many key myelin and axonal cytoskeletal proteins are calpain substrates [201]. Post-mortem analyses of MS patients have also shown a marked increase in calpain expression and activity within demyelinated lesions, without any corresponding alterations in calpastatin [202]. Calpain degradation products, including α-spectrin, have also been found in the brains of individuals with MS [203]. The heightened calpain signal is localized to reactive astrocytes, infiltrating T cells, and activated macrophages/microglia within and surrounding demyelinated plaques [204,205]. It is worth noting that activated T lymphocytes not only express calpain, but also secrete it extracellularly, further contributing to a massive proteolysis of myelin and membrane proteins on oligodendrocytes [206].

Findings from animal models reinforce this mechanism. In experimental autoimmune encephalomyelitis (EAE), which recapitulates many features of MS, calpain expression and activity rise sharply at the peak of inflammation [207]. As in human tissue, calpain expression is increased in infiltrating immune cells, glia, and neurons within demyelinated lesions [12,208,209,210]. In MS, calpain is considered as a key driver of myelin degradation as it can degrade all main myelin proteins [206,211]. A protective effect has been observed in EAE models through therapeutic inhibition of calpain. Calpeptin, for instance, has the potential to serve as a pharmacological treatment for multiple sclerosis by decreasing gliosis, myelin loss, and cell death [212]. Likewise, the behavioral outcomes were enhanced and inflammation and demyelination were prevented by the oral administration of a BBB permeable calpain inhibitor one week after EAE induction [213]. Overall, the correlation between neurodegeneration in MS and calpain overactivation is well-documented.

## 5. Calpain Fingerprints as Biomarkers in Spinal Cord Disorders

A central challenge in neuroscience and clinical research is the pursuit of biomarkers that are both biologically meaningful and non-invasive, capable of reliably monitoring the course of a medical condition. Current diagnostic, prognostic, and monitoring instruments are frequently hindered by low sensitivity, limited specificity, or poor predictive value in spinal cord disorders. An opportunity to identify novel biomarkers with clinical relevance is now available as a result of a more comprehensive understanding of the molecular events that underlie these pathologies. In this context, protein fragments derived from calpain have shown particular promise.

### 5.1. Spectrin and GFAP Breakdown Fragments

Spectrin breakdown products (SBDPs) represent the initial calpain-specific biomarkers suggested for the assessment of CNS trauma. In patients with severe traumatic brain injury (TBI), a condition that significantly overlaps with the pathophysiology of SCI, SBDPs are detectable in the cerebrospinal fluid (CSF) within hours post-injury [214]. Initial CSF levels of SBDP145 correlate positively with the severity of injury and mortality in TBI. In a classic rat contusion study, SBDPs were also observed to accumulate in injured spinal cord tissue as early as 15 min post-impact, reaching a peak at 2 h [96] (see Figure 2B and Figure 3). Subsequent research has demonstrated that SBDP145 fragments are detected in the CSF immediately following an injury [104]. Levels increase significantly within 4 h, remain elevated at 24 h, decline near baseline by 48 h, and are undetectable by day 7. The peak concentration of these fragments was significantly correlated with the severity of injury. Translational evidence reflects this pattern. In the sole prospective series conducted thus far, serial CSF samples from patients with acute traumatic SCI indicated that SBDP150/145 levels were significantly elevated from day 1 to day 4 in comparison to neurologically normal controls, but subsequently decreased and approached baseline levels by day 5 [215]. Peak values correlated with injury severity, indicating that patients with complete injuries exhibited significantly higher SBDP levels compared to those with incomplete injuries. Therefore, the data that has been collected in humans thus far indicates that calpain substrates and their fragments have the potential to serve as biomarkers and contain clinical information.

In parallel, GFAP and its breakdown products have gained attention as complementary biomarkers. GFAP offers a level of cell-type and CNS specificity that spectrin does not, as it is almost exclusively expressed in astrocytes. After SCI, GFAP-BDPs accumulate in spinal tissue for up to 7 days in a severity-dependent manner. However, they have been detected in the CSF at 4- and 24 h post-injury, but not at 7 days, showing a transitory presence in the acute phase [104] (see Figure 3). Clinical investigations have also reported that CSF GFAP levels are significantly elevated in SCI patients, with a positive correlation to the severity of the injury [216].

Despite their potential, spectrin and GFAP fragments have not become routine biomarkers for SCI, presumably due to a variety of impediments. The first limitation is that they are not specific. During apoptosis, calpain and caspase-3 cleave αII-spectrin and GFAP [217,218], resulting in overlapping fragments that obscure the distinction between necrotic and apoptotic processes. The second point is that SBDPs and GFAP are transient. As we just saw, they rise rapidly after injury and then decline within days, making them not useful for monitoring subacute or chronic complications, such as spasticity. Third, spectrin is not specific to the CNS. It is detectable in red blood cells and peripheral tissues [219], meaning that elevated levels may be the consequence of systemic injury or hemolysis, particularly in polytrauma patients. Despite being CNS-specific, GFAP is a separate but complementary indicator of tissue damage, as it is indicative of astroglial injury rather than neuronal injury. These limitations emphasize the importance of next-generation biomarkers that are neuron-specific, calpain-selective, and present beyond the acute phase.

### 5.2. Next-Generation Calpain-Mediated Biomarkers: Toward Disease Specificity

#### 5.2.1. Traumatic Injury and Spasticity

Calpain-mediated sodium channel fragments have emerged as a promising biomarker candidate. They are specific to neurons, absent from peripheral blood under normal conditions, and cleaved by calpains but not caspases, which adds to their specificity [220]. In preclinical models of SCI, Nav channel fragments persist in spinal tissue well into the chronic phase [68,81,122]. Although their presence in blood has not yet been confirmed, we hypothesize that they could reach the circulation through extracellular vesicles or as a consequence of BBB disruption.

Other products of calpain activity, such as KCC2 fragments, also hold potential as blood-based biomarkers. In a prenatal hypoxia-ischemia rat model, serum KCC2 levels reflected the extent of cerebral injury [221]. As KCC2 is involved in chloride homeostasis and is downregulated in SCI-related spasticity [90], these fragments may represent a neuron-specific, calpain-sensitive marker in the subacute phase.

To test the clinical relevance of such neuronal calpain-mediated proteolytic fragments, we recently launched a multicenter prospective SpasT-SCI-T clinical trial [222]. This study will follow blood Na_v_ fragment levels in patients with acute SCI or TBI over 6 months, mapping their trajectory against neurological outcomes, including spasticity onset and severity. Currently clinicians have no reliable way to predict these outcomes; the management of spasticity is too often based on a “wait and see” approach. Detecting robust biomarkers in the early post-injury period could allow a shift toward anticipatory, proactive care, offering the possibility of improving long-term outcomes. In addition to their diagnostic value, such circulating calpain-derived fragments may also serve as indirect readouts of treatment efficacy. Anti-inflammatory therapies that limit edema and gliosis could, in principle, attenuate the release of these proteolytic products into plasma. Future work should assess whether monitoring calpain-related fragments in blood can provide a practical biomarker strategy for evaluating the success of anti-inflammatory interventions in SCI patients.

#### 5.2.2. Amyotrophic Lateral Sclerosis

Neurofilament light chain (NfL) is one of the most established cytoskeletal protein biomarkers in ALS. Elevated levels are consistently found in both blood and CSF, and these correlates closely with disease progression [223,224,225]. As noted earlier, neurofilaments are direct substrates of calpains [226,227], raising the possibility that calpain-mediated proteolysis contributes, at least in part, to the release of NfL fragments into the extracellular space. Calpain also generates other products that may enrich the biomarker profile of ALS; for example, TDP-43 fragments have been detected in patient CSF [187,228]. For now, NfL remains the most robust biomarker validated for clinical use. However, new avenues are emerging. Biomarkers derived from sodium channels, particularly Nav1.6 fragments, may prove valuable, not only for ALS but also for comparative studies with SCI pathology. Another promising candidate is KCC2. Recent reports show its presence in the CSF and plasma of patients and in mutant mouse models [229], suggesting that calpain-mediated KCC2 fragments are released into biofluids as the disease progresses.

#### 5.2.3. Multiple Sclerosis

In the EAE model of MS, calpain expression and activity in the spinal cord is markedly upregulated at the very onset of symptoms, just as myelin basic protein (MBP) begins to undergo proteolysis, but before demyelination is clearly evident [212,230]. Pharmacological or genetic inhibition of calpain in EAE models reduces MBP degradation, inflammatory cell infiltration and neurologic deficits, pointing strongly to its role in myelin loss [207]. In clinical settings, the biochemical readout of active demyelination has been the appearance of MBP degradation products in CSF. The initial longitudinal studies showed that MBP levels increase during relapses and decrease during remission, thereby establishing a quantifiable indicator of disease activity over time [231]. Post-mortem studies support this link, revealing MS plaques rich in calpain-1 and calpain-2, along with spectrin and MBP fragments generated by calpain cleavage [232].

Taken together, these data highlight the broad potential of calpain-generated protein fragments as molecular fingerprints of spinal cord pathologies. From trauma to neurodegeneration and demyelination diseases, calpain leaves a proteolytic trail in biofluids that often mirrors disease severity or progression. First-generation markers like spectrin fragments got us started, next-generation markers like sodium channel and KCC2 fragments may be more CNS-specific and relevant beyond the acute phase. Integrating these biomarkers into the clinic could not only improve diagnosis but also move us towards earlier, more personalized treatments for spinal cord disorders.

## 6. Disarming Calpain: Genetic and Pharmacological Tools for Spinal Cord Repair

Given the consistent upregulation of calpain in SCI, neuropathic pain and neurodegenerative diseases, a range of therapeutic strategies has been explored to suppress its activity. These include small-molecule inhibitors, gene therapies designed to boost the endogenous inhibitor calpastatin, and, more recently, antisense oligonucleotides directed against specific calpain isoforms. The following sections review the progress achieved with these approaches, the challenges that remain, and ongoing efforts to bring them into clinical application.

### 6.1. Pharmacological Approaches

Numerous calpain inhibitors have been evaluated in experimental SCI models. In acute injury, broad-spectrum drugs, including AK295, MDL-28170, and calpeptin, have demonstrated neuroprotective effects, which include the reduction in tissue damage and the enhancement of functional outcomes [233,234,235,236]. As illustrated in Figure 5, the systemic administration of MDL-28170 following SCI reduces calpain activity in spinal motoneurons, thereby preserving key substrates such as KCC2 and Nav1.6. Interestingly, even short-term administration in the chronic phase can have lasting benefits; in spastic rats, a 10-day treatment reduced spasticity, with benefits still present one month after the therapy ended [81]. These results indicate that the trajectory of chronic complications can be disrupted by the inhibition of calpain, which protects ion channels and receptors from proteolytic damage. In models of neuropathic pain, the immediate administration of drugs after nerve injury prevented pain hypersensitivity by maintaining KCC2 levels and limiting cytokine release [166]. However, its efficacy in established neuropathic pain remains unclear [237].

Although no pharmacological studies have yet tested calpain inhibition in ALS models, benefits have been found in EAE, a model of MS. Membrane-permeable calpain inhibitors produced a dose-dependent improvement of clinical signs, attesting to the calpain involvement in the development of paralysis and tissue damage in MS a [212]. Similar effects were seen with SNJ-1945, a novel inhibitor, which reduced paralysis scores in EAE mice [238]. All these findings reinforce the idea that calpain might be a good target for MS treatment.

However, significant challenges exist in the clinical application of calpain inhibitors to human therapy. Side effects and selectivity continue to be significant concerns, as current calpain inhibitors also inhibit other cysteine proteases, including cathepsins, which increases the likelihood of off-target effects [239]. Furthermore, a prolonged period of systemic inhibition might interfere with normal physiological processes, making localized delivery approaches more attractive. Therefore, the success of clinical trials will be contingent upon the development of next-generation inhibitors that exhibit even greater isoform specificity. Significantly, a mouse model of repeated concussions has demonstrated that a calpain-2 selective inhibitor can prevent pathological symptoms [240,241].

### 6.2. Gene Therapy

The limitations of systemic pharmacological inhibition have prompted the search for alternative strategies. One promising approach is gene therapy, which has the potential to achieve isoform-selective, cell-specific, and sustained calpain modulation.

#### 6.2.1. Calpastatin Overexpression: Broad-Spectrum Inhibition of CNS Calpains

Calpastatin, the endogenous inhibitor of calpains, is found alongside them in both the cytosol and the membrane [9,242]. It is highly selective for calpain-1 and calpain-2, while leaving tissue-specific calpains, such as those in skeletal muscle, and other proteases unaffected [243]. However, its large size prevents its cell entry. For this reason, and as illustrated in Figure 5, transgenic overexpression of calpastatin has been used as a biological approach to inhibit calpain activity in vivo. Mice with elevated calpastatin levels following spinal cord contusion exhibited reduced calpain activity, diminished neuronal loss, and enhanced locomotor recovery in CNS injury models [244]. In models of ALS, neuron-specific overexpression restored spinal cord calpain activity, averted the proteolytic degradation of critical cytoskeletal proteins, and diminished motor axon degeneration [180,245]. These effects preserved neuromuscular junctions, delayed symptom onset and paralysis, and extended lifespan, providing clear evidence that chronic calpain hyperactivation is a major driver of ALS pathogenesis. By contrast, direct studies of calpastatin overexpression in neuropathic pain or MS models have yet to be reported.

Despite its promise, calpastatin has notable limitations. It is itself a substrate for calpain [98,246,247]. Furthermore, calpastatin does not discriminate between calpain-1 and calpain-2. As these isoforms can play distinct and occasionally opposing functions, broad inhibition may not be the most effective approach. This has driven the development of more targeted approaches to specific calpain isoforms.

#### 6.2.2. Isoform-Specific Knockdown Approaches

Targeted gene therapy offers a means to selectively silence the expression of individual calpain isoforms. As illustrated in Figure 5, our team recently applied this strategy to knockdown calpain-1 in a model of chronic SCI-induced spasticity [93]. In this study, adult rats with SCI received lumbar intrathecal injections of AAV6-shRNA. Calpain-1 knockdown occurred primarily in lumbar motoneurons.

The expression of calpain-1 in transduced motoneurons was reduced by 70%, which resulted in a reduction in spinal hyperexcitability. Consequently, the progression of spasticity to more severe stages was subsequently prevented. Compared to controls, rodents that were treated with calpain depletion exhibited a reduced incidence of hyperreflexia, muscle spasms, and clonus. The cellular-level calpain suppression resulted in the restoration of KCC2 levels in motoneurons, suggesting that this critical regulator of neuronal inhibition was rescued from calpain-mediated proteolysis. As a result of the preservation of KCC2, spasticity was reduced, and motoneurons became less hyperexcitable. The proof-of-concept of calpain-1 as a therapeutic target in chronic SCI is provided by this motor improvement. The results of these studies are in agreement with those of acute SCI studies, which also demonstrated a substantial improvement in locomotor function and tissue sparing following the intraspinal injection of lentiviral vectors encoding calpain-1 shRNA in rats prior to contusion [109]. Therefore, the downregulation of calpain-1 represents an efficient therapeutic approach for both acute and chronic SCI.

Isoform-specific approaches may also be relevant for calpain-2, particularly in chronic pain, where neuroinflammation plays a major role. Following motor nerve injury, calpain-2 is upregulated in DRG neurons and promotes the expression of pro-inflammatory cytokines such as TNF-α and IL-6, both of which contribute to pain hypersensitivity [166,167,248]. Calpain-2 inhibition alleviates allodynia and reduces TNF-α and IL-6 levels, whereas exogenous calpain-2 expression in intact animals induces cytokine upregulation and pain behaviors. These findings emphasize the critical function of calpain-2 in the development of neuropathic pain and neuroinflammation.

No isoform-specific genetic approaches have yet been tested for calpain-1 or calpain-2 in ALS or MS models. The most advanced concept in this field is a therapy for ALS developed by Amylyx Pharmaceuticals: an antisense oligonucleotide (ASO) designated AMX0114 that targets calpain-2 mRNA for degradation. Late in 2024, the company initiated the Phase 1 LUMINA trial of AMX0114 in ALS patients (NCT06665165). Delivered intrathecally, this ASO is directed against calpain-2 and, if effective, would represent the first isoform-specific calpain therapy to reach clinical application. Such a success would pave the way for similar strategies, for example, an ASO against calpain-1 in the treatment of spasticity.

Looking ahead, gene therapies for SCI and ALS will benefit from tools that more precisely target motoneurons. Retrograde, muscle-tropic viral vectors, next-generation AAV serotypes that are designed for efficient uptake at the neuromuscular junction, are one promising approach. This would allow for the selective transduction of motoneurons that innervate spastic or dysfunctional muscles, while simultaneously sparing healthy ones. Our most recent research, as illustrated in Figure 5, resulted in the widespread, high-efficiency transduction of motoneuron pools, as well as the robust and specific silencing of target genes, through the intramuscular delivery of retrograde AAV2 vectors [132]. This targeted delivery has several advantages: it minimizes off-target effects, preserves normal motor function and allows for region- and circuit-specific intervention. This level of precision would greatly increase the safety and efficacy of gene therapies and provide tailored treatments, reducing systemic exposure. In the future, retrograde musculotropic vectors might become the state-of-the-art for high-precision targeted treatments of motoneuron disorders like spasticity after SCI and ALS.

## 7. Conclusions

In conclusion, the present review shows the dual nature of calpains in the spinal cord. Under physiological conditions, they are essential regulators of adaptive processes such as synaptic plasticity, inhibitory tuning, and the modulation of intrinsic excitability. When abnormally activated, they become central drivers of pathology. In the spinal cord, both acute and chronic hyperactivation of calpains has emerged as a major factor in the development of complications, including spasticity and neuropathic pain after SCI, as well as in neurodegenerative diseases such as ALS and MS. This excessive activity promotes the proteolytic degradation of critical neuronal and glial substrates, disturbing excitability, inhibitory control, and overall network organization.

Besides their pathogenic function, calpains are also a source of promising tools for diagnosis and therapy. As biomarkers, specific protein fragments produced by their activity are beginning to be identified as a potential addition to current clinical and imaging methods for the monitoring of spinal pathologies. From a therapeutic perspective, proof-of-concept efficacy in preclinical models demonstrates that both pharmacological inhibition and isoform-specific gene strategies now provide viable means of modulating calpain activity.

However, their clinical translation remains challenging. Most candidate biomarkers still lack validation in large patient cohorts, their specificity can be confounded by other forms of injury, and their temporal profile often limits their utility. Likewise, pharmacological inhibitors suffer from limited selectivity and potential off-target effects, while efficient BBB penetration remains a major hurdle. Gene therapy approaches, though promising, are still at an early experimental stage.

Altogether, although calpain-related biomarkers and therapies hold significant translational promise, their implementation in clinical practice will depend on overcoming these challenges. Addressing issues of specificity, delivery, and large-scale validation will be essential to transform current experimental advances into effective tools for patient care. Combining future biomarker strategies with precision interventions could eventually move care for SCI and related diseases from reactive, symptom-driven management to mechanism-guided, preventive medicine.

## Figures and Tables

**Figure 1 cells-14-01503-f001:**
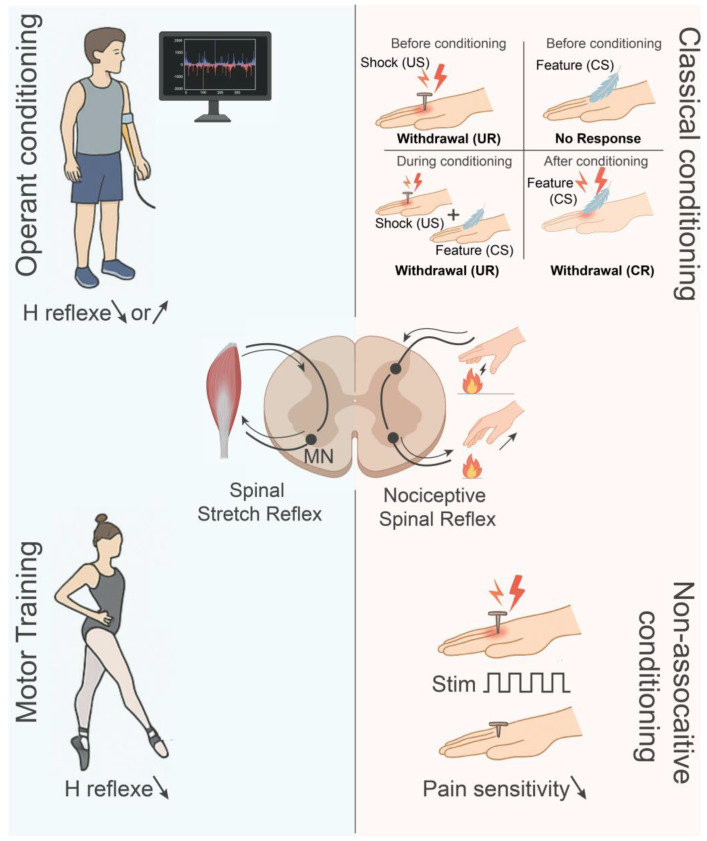
Forms and paradigms of spinal plasticity. The adult spinal cord supports multiple forms of learning, as shown here by key paradigms. Diagrams of the spinal cord indicate the neural circuitry for the stretch reflex or H-reflex (**left**, blue) and nociceptive reflex (**right**, peach) for each paradigm. **Top left**: Operant conditioning of the H-reflex. Subjects (human or animal) learn through feedback and reward to increase or decrease reflex amplitude and obtain persistent changes in synapses and motoneurons. **Top right**: Classical conditioning of spinal reflexes. The repeated pairing of a neutral stimulus (CS, feather) with a noxious stimulus (US, shock) produces associative plasticity. Before conditioning, the US alone elicits an unconditioned response (UR, withdrawal), while the CS alone evokes no response. During conditioning, CS and US are paired, eliciting the UR. After conditioning, the CS alone triggers a conditioned response (CR, withdrawal). **Bottom right**: Non-associative conditioning, including habituation and sensitization, derives from the repeated exposure to the same stimulus and generates a progressive decrease (habituation) or increase (sensitization) in reflex or pain pathways. Symbols: arrows = signaling flow or reflex change; ↑ or ↓ = increase or decrease; MN = motoneurons.

**Figure 2 cells-14-01503-f002:**
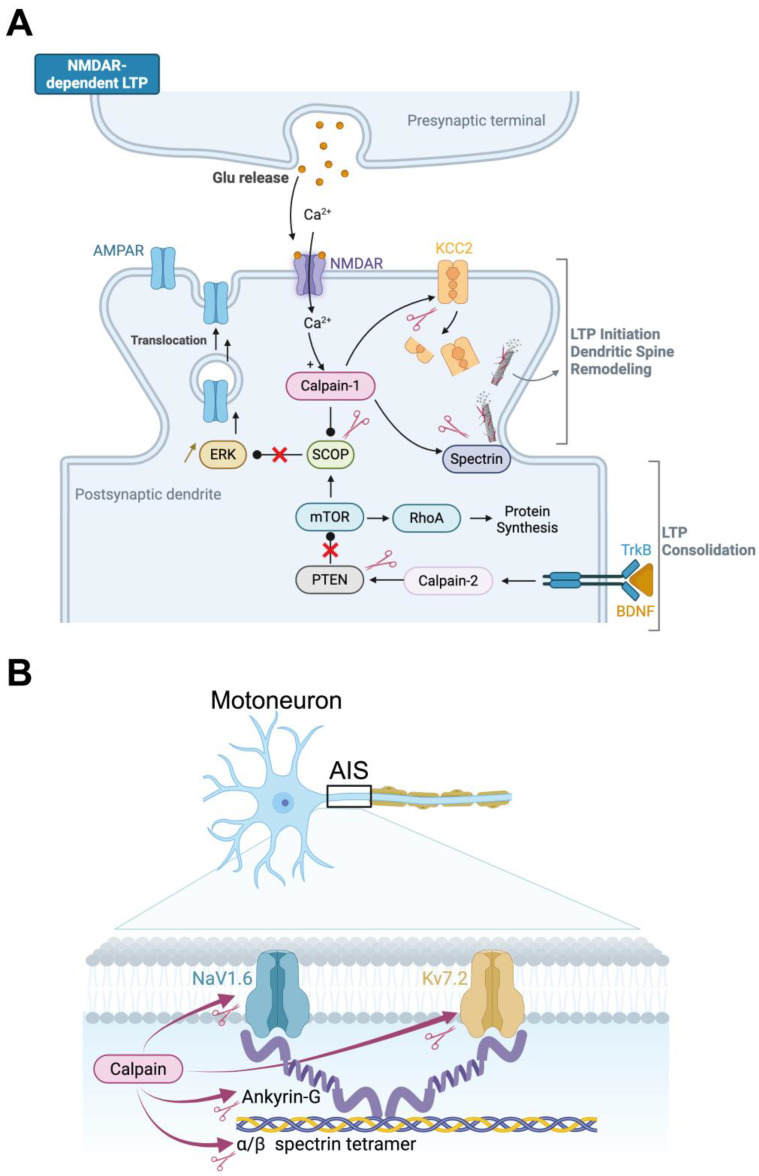
Calpain-dependent control of synaptic and intrinsic neuronal plasticity. (**A**) Calpain as a molecular effector of NMDAR- and BDNF-dependent long-term synaptic plasticity. Activation of NMDA receptors (NMDARs) by glutamate release triggers Ca^2+^ influx into the postsynaptic dendrite, leading to calpain-1 activation. Once active, calpain-1 facilitates the initiation phase of long-term potentiation (LTP) by two mechanisms: (1) cleavage of cytoskeletal proteins, such as spectrin, which facilitates dendritic spine remodeling and insertion of AMPA receptors (AMPARs) and (2) proteolytic degradation of regulatory proteins like SCOP, thereby lifting inhibition of ERK signaling and further strengthening synaptic transmission. Calpain-1 also acts on the KCC2 cotransporter, reducing inhibitory drive and shifting the balance toward excitation. Meanwhile the cell releases BDNF in response to activity which binds to TrkB receptors and activates calpain-2. Calpain-2 cleaves the negative regulator of mTOR, PTEN, and thus promotes mTOR-dependent local protein synthesis which is required for the late consolidation phase of LTP. These findings suggest that calpain-2 links neurotrophin signaling to the local structural and translational mechanisms required for long-term plasticity. (**B**) Calpain-dependent remodeling of the axon initial segment (AIS) and intrinsic excitability. The AIS, the site where action potentials are initiated, depends on the anchoring of Nav1.6 and Kv7.2 channels to scaffold proteins such as ankyrin-G and spectrin. Calpains can cleave both these structural elements and the channels themselves. This proteolysis leads to AIS disassembly, redistribution or loss of ion channels, and direct alterations in their biophysical properties. In particular, cleavage increases the persistent sodium current (I_NaP_) and decreases the M current (I_M_), changes that enhance neuronal excitability and support activity-dependent adjustments in intrinsic firing properties. Symbols: black arrows = signaling flow; brown arrow = ERK activation; black line ending with a filled circle = inhibition; same line crossed with a red X = relief of inhibition; scissors = proteolysis.

**Figure 3 cells-14-01503-f003:**
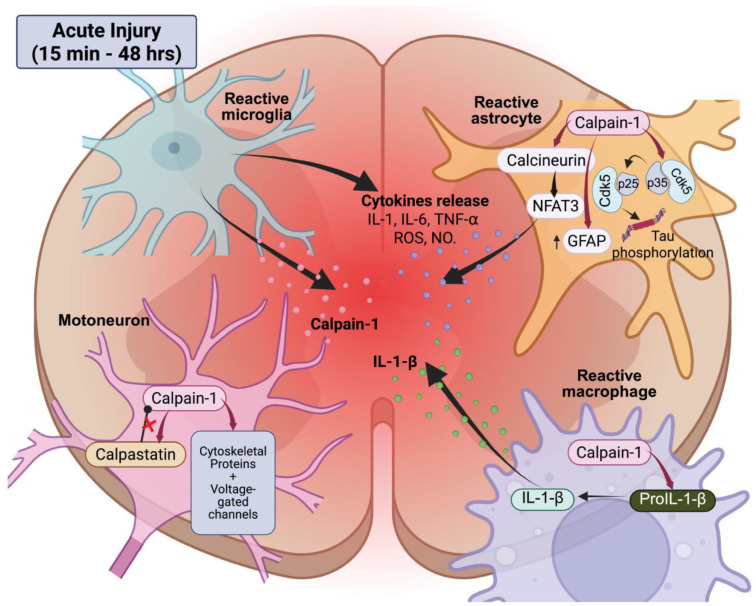
Acute calpain activation and neuroinflammatory responses in the spinal cord after injury. Within minutes of a spinal cord injury (SCI), calpain-1 becomes rapidly activated in multiple cell types within the lesion core and surrounding segments. In motoneurons, this hyperactivation, worsened by the cleavage of its endogenous inhibitor calpastatin, degrades cytoskeletal and membrane proteins, compromising cellular architecture and excitability. Calpain-1 in astrocytes interacts with calcineurin to produce p25, hence maintaining Cdk5 activity, which subsequently phosphorylates tau and promotes astrocyte activation. Calpain-1 also cleaves and upregulates GFAP, a hallmark of reactive astrogliosis. Reactive astrocytes release pro-inflammatory cytokines, IL-1β, IL-6, TNF-α and reactive oxygen and nitrogen species (ROS, NO). In macrophages, calpain-1 converts pro-IL-1β to its active form, which in turn enhances inflammation. Pro-inflammatory cytokines and extracellular calpains are also released by microglia, which contribute to perpetuating a cycle of neurotoxicity. In sum, the synergistic effects of calpain activation, calpastatin depletion, and neuroimmune signaling provide a permissive environment for sustained calpain activity. Symbols: red arrows = calpain-mediated proteolysis; black line ending with a filled circle = inhibition; same line crossed with a red X = relief of inhibition.

**Figure 4 cells-14-01503-f004:**
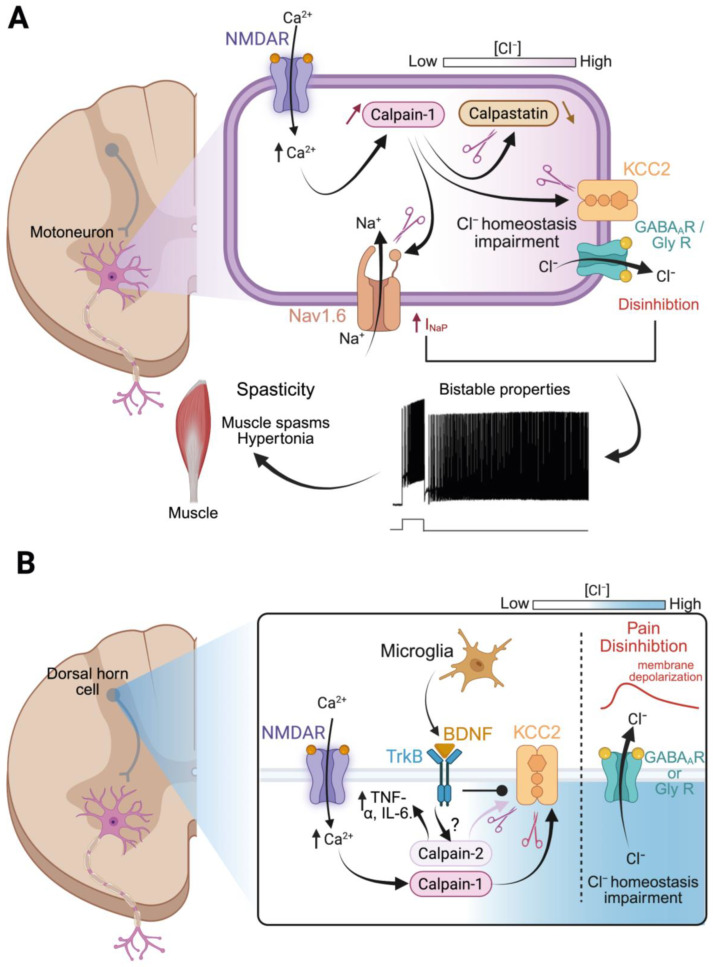
Calpain-mediated mechanisms underlying chronic spasticity and neuropathic pain after spinal cord injury (SCI). (**A**) In motoneurons below a chronic SCI, sustained calpain-1 activation, triggered by increased intracellular calcium, cleaves the inactivation gate of voltage-gated sodium (Na_v_) channels, resulting in persistent sodium currents (I_NaP_) and bistable firing. Calpain-1 also mediates the proteolytic cleavage of the potassium-chloride cotransporter KCC2, which leads to chloride homeostasis disruption and a reduction in inhibitory synaptic transmission. Finally, calpain-1 cleaves its own endogenous inhibitor, calpastatin, thus further increasing calpain activity. This cascade of events leads to motoneuron hyperexcitability, exaggerated reflex activity and spasticity. (**B**) In the dorsal horn, injury-induced microglial activation releases BDNF which activates TrkB signaling and upregulates calpain-1/2, further promoting KCC2 cleavage and the release pro-inflammatory cytokines (TNF-α, IL-6). This disrupts the chloride gradient, resulting in disinhibition and central sensitization, a key component of chronic neuropathic pain. Symbols: black arrows = signaling flow; black line ending with a filled circle = inhibition; scissors = proteolysis; “?” = hypothetical pathway.

**Figure 5 cells-14-01503-f005:**
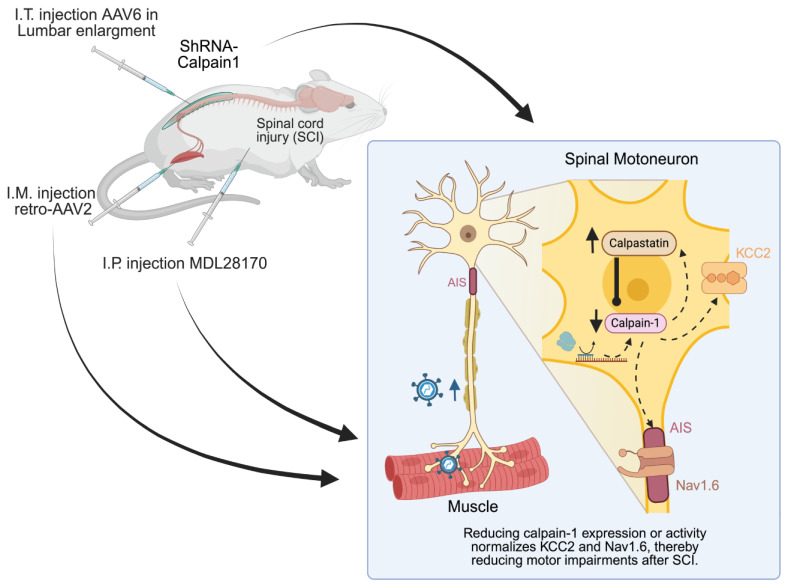
Pharmacological and genetic strategies targeting calpain activity for spinal cord repair. (**Left**) Schematic of in vivo interventions to reduce calpain-1 activity in rodent models of SCI, including intrathecal (I.T.) injection at the lumbar enlargement of AAV6 vectors encoding calpain-1-directed shRNA, intramuscular (I.M.) injection of retrograde AAV2 vectors, systemic (I.P.) injection of the broad-spectrum calpain inhibitor MDL28170 and genetic overexpression of calpastatin, the endogenous calpain inhibitor. (**Right**) These pharmacological and genetic approaches converge on the motoneuron where the reduction in calpain-1 expression or activity preserves KCC2 expression and chloride homeostasis, maintains calpastatin levels and protects Na_v_1.6 channel function at the axon initial segment (AIS). Together, this reduces motoneuron hyperexcitability, supports inhibitory tone and ameliorates motor dysfunctions such as spasticity after SCI. **Symbols:** ↑ or ↓ = increase or decrease; black lines ending with a filled circle = inhibition; dashed arrows = reduced downstream effects of calpain-1 on its targets; blue arrow = retrograde transport of viral vectors in motoneuron axons.

## Data Availability

No new data were created or analyzed in this study.

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
