# Peer review of "Calpains at the Crossroads of Spinal Cord Physiology, Plasticity, and Pathology"

_cells, 2025, doi:10.3390/cells14191503_

Round 1

Reviewer 1 Report

Comments and Suggestions for Authors

Brocard and Dingu provide a thorough review of the role of calpains in the pathophysiology of traumatic and neurodegenerative spinal cord disorders. In addition, they discuss the use of calpain-related products as biomarkers for damage/disease progression and the therapeutic potential of targeting calpain activity. The review is well organized and the material is written at a level that should be accessible to a broad audience. I particularly liked how they linked calpains to research on BDNF and KCC2. Overall, I believe that the paper is in excellent shape and will have a high impact.

There were some minor issues that I felt require attention. One concerns the figures. While these are logically laid out, I suspect that many of the embedded components will become unreadable when reproduced. This was particularly true for Figures 3 and 4 (and a moderate concern for Figure 5). Throughout the authors need to use font sizes that remain readable (without enlarging the figures).

In Figure 1, I believe that the authors mean (line 110) that the spinal cord “supports multiple forms of learning” rather than “many forms of plasticity” (alternative forms of learning appear to depend upon common components [e.g., NMDAR-dependent plasticity]). I also found the illustration for classical conditioning confusing. Part of the problem is that the elements and their order are hard to decipher. More generally, it was not clear what was serving as the CS, US, CR or UR. There are may published illustration of this paradigm that the authors could borrow from to construct a clearer figure.

There were also a number of places where the text could be edited to improve clarity and/or flow. This includes lines:

63-65 (“this points out on a”);

164-165 (“So calpain- 2 links” [“So” could be changed to “The findings suggest that”]);

411-413 (“we reasonably assume that they might be”);

546 (“this” [unclear referent]);

594-594 (the authors note the “second” and “third” points, but not the first [likely implied by “initially”]);

615 (the material starting at “To” signals a new concept and seemingly, a new paragraph);

630 (“generate” should be plural).

In addition, the authors sometimes use “since” (which should only be used in reference to time) when they mean “because” (lines 411, 615).

Author Response

Reviewer 1: We thank the reviewer for the very positive evaluation of our manuscript and for the constructive suggestions, which have helped us improve both the clarity and readability of the review. We have addressed all comments point by point below.

Comment 1. Figures readability (Figures 3–5)

Reviewer comment: “…many of the embedded components will become unreadable when reproduced. This was particularly true for Figures 3 and 4 (and a moderate concern for Figure 5). Throughout the authors need to use font sizes that remain readable…”

Response 1: We agree and have increased the font size throughout Figures 3, 4, and 5 to improve legibility. We also simplified several embedded components to ensure that all elements remain readable when reproduced.

Comment 2. Figure 1 “forms of learning” and classical conditioning illustration

Reviewer comment: “…line 110, the spinal cord ‘supports multiple forms of learning’ rather than ‘many forms of plasticity’… The illustration for classical conditioning is confusing…”

Response 2: We thank the reviewer for these valuable suggestions. In response, we have revised the text in the body of the manuscript (line 110) to read “the spinal cord supports multiple forms of learning” instead of “many forms of plasticity.” In addition, we have redesigned Figure 1 and updated its legend. The panel illustrating classical conditioning now explicitly identifies the conditioned stimulus (CS), unconditioned stimulus (US), unconditioned response (UR), and conditioned response (CR). The sequence of events (before conditioning, during conditioning, after conditioning) is clearly labeled, and the layout has been simplified to improve readability. The revised legend (Page 4 lines 115-119) also describes these components explicitly to facilitate understanding.

Comment 3. Textual clarity and flow:: “…lines 63–65; 164–165; 411–413; 546; 594–595; 615; 630…”

Response 3: We carefully revised these passages for clarity and flow:

  • Line 63–65: corrected to “this highlights” instead of “this points out on a”.
  • Line 164–165: “So calpain-2 links” replaced with “These findings suggest that calpain-2 links”.
  • Line 422: “we reasonably assume that they might be” simplified to “we assume that they may be”.
  • Line 556: clarified the referent of “this”.
  • Lines 603: rephrased to explicitly mention the “first” point before the “second” and “third”.
  • Line 615: separated into a new paragraph as suggested.
  • Line 630: corrected “generate” by “generates”.

Comment 4. Use of “since” vs. “because”: We have corrected these instances by replacing “since” with “because” or “as” to ensure proper usage.

Reviewer 2 Report

Comments and Suggestions for Authors

GENERAL COMMENTS:

  • The authors need to review recent literature on the pathogenesis of the disease initiated by SCI and provide a more accurate context for the role of calpains in this very severe, destructive and very protracted disease
  • The response of spinal cord, primarily astrogliosis, to SCI injury and inflammatory disease needs to be discussed with an attempt to provide this process as the proper setting for interpretation of the role of calpains. In SCI neurons are innocent victims and do not appear to participate actively in anti-inflammatory = neuroprotective tissue response.
  • Vasogenic edema in the non-destroyed portion of peri-lesional spinal cord is the therapeutic target for anti-inflammatory therapies in animal models of SCI and very recently in treated SCI patients who show improvement in motor deficits (in preparation). Is there a role for calpains in damage to blood spinal cord barrier?

What would be the authors’ suggestion for detection of calpains either directly or indirectly in the blood plasma of SCI animal models or human patients to help monitor the success of anti-inflammatory treatment?

Author Response

Reviewer 2:

Comments 1: “The authors need to review recent literature on the pathogenesis of the disease initiated by SCI and provide a more accurate context for the role of calpains in this very severe, destructive and very protracted disease. The response of spinal cord, primarily astrogliosis, to SCI injury and inflammatory disease needs to be discussed with an attempt to provide this process as the proper setting for interpretation of the role of calpains. In SCI neurons are innocent victims and do not appear to participate actively in anti-inflammatory = neuroprotective tissue response.”

Response 1: We agree with the reviewer that astrogliosis and inflammation dominate the early tissue response to SCI, while neurons are largely passive targets of these cascades. These aspects were already addressed in the manuscript, notably in Section 3.1.1 and Figure 3, as well as in Section 3.2.2 (neuroimmune interactions in pain) and Section 3.3.2 (MS). To clarify this point further, we have added a sentence in the introduction of the SCI section (page 9 lines 268-269) and emphasized at the end of Section 3.1.1 (page 10, lines 297-299) that, although gliosis represents an important calpain-dependent feature of the injured microenvironment, it is ultimately the neuronal consequences of this glial response that drive long-term dysfunction.

Comments 2: “Vasogenic edema in the non-destroyed portion of peri-lesional spinal cord is the therapeutic target for anti-inflammatory therapies in animal models of SCI and very recently in treated SCI patients who show improvement in motor deficits (in preparation). Is there a role for calpains in damage to blood spinal cord barrier?”

Response 2: We thank the reviewer for raising this point. Direct evidence for a role of calpains in blood–spinal cord barrier (BSCB) dysfunction after SCI is limited. However, studies in brain trauma and ischemia have demonstrated that calpains contribute to blood–brain barrier breakdown through cleavage of cytoskeletal and tight junction proteins, thereby promoting vascular leakage and edema formation (Alluri et al., 2016; Tao et al., 2017). By analogy, we propose that similar mechanisms may also occur at the BSCB after SCI, potentially exacerbating vasogenic edema in peri-lesional regions. To acknowledge this possibility, we have added a short paragraph in Section 3.1.1 (page 10 lines 303-308) highlighting calpains as potential mediators of vascular barrier dysfunction and edema formation, and we have included two additional references (Alluri et al., 2016; Tao et al., 2017) in the bibliography.

Comments 3: “What would be the authors’ suggestion for detection of calpains either directly or indirectly in the blood plasma of SCI animal models or human patients to help monitor the success of anti-inflammatory treatment?”

Response 3: We thank the reviewer for this suggestion. Our manuscript already discusses circulating calpain-derived protein fragments (such as sodium channel or spectrin breakdown products) as promising biomarker candidates (Section 4.2.1). To make their therapeutic relevance more explicit, we have now added at the end (page 20 lines 633-637) of the section sentences highlighting that these fragments could serve not only as diagnostic indicators but also as indirect readouts of treatment efficacy. In particular, anti-inflammatory interventions that reduce edema and gliosis might attenuate the release of calpain-dependent fragments into plasma, and monitoring such biomarkers could therefore help evaluate the success of these therapies.

Reviewer 3 Report

Comments and Suggestions for Authors

In this review the authors present the molecular mechanisms of calpains, their cellular localization, and their role in spinal cord injury. The review highlights that calpains play multiple roles in the central nervous system, such as in synaptic plasticity of spinal cord, spasticity, and the development of neuropathic pain. The number of references is high and up to date, which demonstrates that the manuscript is built on a strong scientific foundation.

It is important to emphasize that spinal plasticity is interpreted from the perspective of calpain activity, which is significant because research in this field generally focuses on cortical processes.
Another strength of the article is that it directly connects molecular mechanisms with clinical symptoms and biomarkers.

  • The presentation of calpain-mediated ion channel fragments (Nav1.6, KCC2) as potential biomarkers is particularly valuable and consistent with current translational trends.
  • It is also an advantage that the authors discuss not only pharmacological inhibitors but also gene therapy approaches (calpastatin overexpression, antisense oligonucleotides).

The main weakness lies in the clinical applicability, since currently there are few human data to support that these biomarkers or therapeutic strategies can truly be implemented in practice. The manuscript is at times overly optimistic about translational prospects; therefore, it would be important to highlight the limitations (e.g., specificity, side effects, blood–brain barrier penetration).

Overall, the manuscript is a high-quality, well-structured, and comprehensive review that contributes to the understanding of the roles of calpains. The other strength of the MS is that the authors integrate perspectives of plasticity, pathology, and biomarkers/therapy.

Author Response

Reviewer 3 : We thank Reviewer 3 for their constructive feedback and positive evaluation. We are pleased that they valued the comprehensive scope of the manuscript, its novel perspective on spinal plasticity through calpain activity, and the links drawn to biomarkers and therapies.

Comment 1: “The main weakness lies in the clinical applicability, since currently there are few human data to support that these biomarkers or therapeutic strategies can truly be implemented in practice. The manuscript is at times overly optimistic about translational prospects; therefore, it would be important to highlight the limitations (e.g., specificity, side effects, blood–brain barrier penetration).”

Response 1:  We thank the reviewer for this important remark. While some of these limitations were already mentioned in the original version of the manuscript, we agree that they needed to be emphasized more clearly. Rather than adding repetitive statements throughout the biomarker and therapeutic sections, we have revised the conclusion (pages 24-25 lines 798-808) to provide a more balanced view of translational prospects. The revised conclusion now explicitly highlights the current limitations of calpain-related biomarkers and therapeutic strategies, including the lack of large-scale validation, issues of specificity, potential off-target effects, and the challenges of BBB penetration. We believe that this addition provides the necessary critical perspective while keeping the manuscript concise and focused.

Round 2

Reviewer 2 Report

Comments and Suggestions for Authors

no comments